# Post-Discharge non-invasive ventilation for hypercapnic respiratory failure: Outcomes in a Rural Cohort

Sunil Sharma[1]*, Robert Stansbury[1,2], Mayuri Mudgal[1], Priyanka Srinivasan[1], Edward Rojas[1], Kassandra K. Olgers[1], Scott Knollinger[3], Bernardo J. Selim[4], Sijin Wen[5]

1 Division of Pulmonary, Critical Care and Sleep Medicine, West Virginia University School of Medicine, Morgantown, West Virginia, 2 Department of Medicine, University of Pittsburgh, Pittsburgh, Pennsylvania, 3 Department of Respiratory Care, Ruby Memorial Hospital, Morgantown, West Virginia, 4 Division of Pulmonary and Critical Care Medicine, Center for Sleep Medicine, Mayo Clinic, Rochester, Minnesota, 5 Department of Epidemiology and Biostatistics, West Virginia University, Morgantown, West Virginia

* sunil.sharma@hsc.wvu.edu

## Abstract

### Rationale

Patients with acute-on-chronic hypercapnic respiratory failure suffer from recurrent readmissions due to acute exacerbations. These patients carry higher readmission and mortality rates compared to the general population.

### Objectives

We aimed to delineate the outcomes of patients discharged on non-invasive ventilation.

### Methods

A cross-sectional study was conducted. One hundred nine patients with acute-on-chronic hypercapnic respiratory failure were evaluated and qualified for non-invasive ventilation at discharge. Adherence data was collected post-discharge and the following outcomes were evaluated: 12-month mortality, 6-month hospital readmission, and emergency department visits in patients who were adherent with non-invasive ventilation therapy versus non-adherent and controls.

### Measurements and main results

Of the 95 patients discharged on non-invasive ventilation, 25 patients (26%) were found to be adherent to post-discharge home non-invasive ventilation. The adherent group had significantly lower 12-month mortality (p=0.022). Survival benefit persisted on multivariate analysis with the Cox regression model adjusting for comorbidities and demographics (HR = 0.05, p=0.04). The study also observed reduced emergency department visits in the non-invasive ventilation-adherent group compared to the non-adherent (3% vs 17%) [p=0.049] and controls (3% vs 25%) [p=0.024].

**Data availability statement:** Data cannot be shared publicly because of patient privacy issue. Data are available from the WVU Institutional Data Access / Ethics Committee, which can be contacted via Wesley Kimble, MPA. Mr. Kimble is an IRB board member with expertise in using clinical data for research, federal regulations, institutional regulations, and bioethics. Mr. Kimble, along with members of his team, act as honest brokers for West Virginia University and he is also the HIPAA Expert Determiner for the institution. Data for this project were provided and managed by Mr. Kimble's team. Contact information: Wesley Kimble, MPA Director, Research Data Analytics West Virginia Clinical and Translational Science Institute (WVU) wkimble1@hsc.wvu.edu.

**Funding:** "Research reported in this publication was supported by the National Institute Of General Medical Sciences of the National Institutes of Health under Award Number 5U54GM104942-08. The content is solely the responsibility of the authors and does not necessarily represent the official views of the National Institutes of Health."

**Competing interests:** SS declares receives grant and support for attending meetings or travel from West Virginia University (WVU) and National Science Foundation; is on speakers bureau of Zoll Respicardia Inc. Grant funding from INARI Medical Inc PEERLESS II trial, NIH RECOVER trial, rEST trial by Zoll Respicardia Inc.; and holds patents with WVU (invention numbers: US2023/0122156) that has been licensed by Premash Inc. SS is consultant for Premash Inc. Dr. Stansbury reports grants from NIH. Dr. Rojas and Dr. Mudgal have no conflicts to declare. Priyanka Srinivasan, Kassandra Olgers, Scott Knollinger, and Wen Sijin also have no conflicts to declare.

**Abbreviations:** OHS: Obesity hypoventilation syndrome; COPD: Chronic obstructive pulmonary disease; RLD: restrictive lung disease.

## Conclusions

Hypercapnic respiratory failure patients discharged home with non-invasive ventilation in the adherent group had significantly lower mortality and emergency department visits. Apart from mask intolerance, low health literacy and transfer to skilled nursing facilities were identified as major reasons for non-adherence.

## Introduction

Acute-on-chronic hypercapnic respiratory failure is a common cause of readmissions and mortality [1,2]. Hypercapnic respiratory failure is a shared end point of several conditions, the most common of these being chronic obstructive pulmonary disease (COPD), obesity hypoventilation syndrome, interstitial lung disease, and neuro-muscular disorders in hospitalized patients[3]. The mortality rate due to hypercapnic respiratory failure is reported to be high, ranging between 13–36% [3–5].

Non-invasive ventilation therapy has significantly improved outcomes on acute-on-chronic hypercapnic patients during hospitalization in terms of decreased mortality and need for invasive ventilation [6–8]. There is also data on the benefit of non-invasive ventilation in patients with stable chronic hypercapnic respiratory failure, observing improved quality of life and possibly improved survival [9–11].

Acute-on-chronic hypercapnic respiratory failure is a common problem in rural West Virginia (WV), as the prevalence of both COPD and obesity in WV is one of the highest in the country [12,13]. These conditions pose a unique challenge to the healthcare system in terms of patient care and healthcare utilization. Additionally, rural Appalachia specifically faces multiple barriers in the utilization of this resource-intensive intervention for the management of chronic hypercapnic respiratory failure. Due to the dearth of both primary care providers and specialists, many patients elect to get health care services in the hospital and emergency department (ED) settings [14]. Despite a growing body of evidence on the utility of non-invasive ventilation in hypercapnic respiratory failure, there is a paucity of data on the benefit of discharging patients with non-invasive ventilation after an acute episode of hypercapnic respiratory failure [15]. Currently, there is also no formal process during hospitalization to determine the need for non-invasive ventilation post-discharge.

This study explores the real-world experience of inpatients with acute-on-chronic hypercapnic respiratory failure who met the criteria for receiving post-discharge home non-invasive ventilation therapy. The study examines the all-cause mortality rate after 12 months of discharge, all-cause ED visits, and all cause hospital readmission rates at 6 months of discharge for patients who adhered to their non-invasive ventilation treatment compared to those who did not. Acute-on-chronic hypercapnic hospitalized patients with either COPD and/or BMI > 40 who were discharged without non-invasive ventilation, despite clinical indication(s) on chart review, formed the control group. In our current study, we aimed to assess the differences in clinical outcomes based on post-discharge home non-invasive ventilation usage. In addition, we aimed to assess barriers to adherence to non-invasive ventilation in rural areas in the context of healthcare disparities.

## Materials and methods

### Study design

This is a single-center cross-sectional retrospective study of patients who were admitted to West Virginia University Hospital (WVUH). WVUH has established a formal sleep medicine

program and a hospital sleep medicine team consultation service, which often gets consulted for hypercapnic respiratory failure and non-invasive ventilation therapy. Consultation for evaluation was prompted by the primary admitting team and was not part of a screening protocol. The patients were often also seen by an inpatient pulmonary consultation team in order to have their in-patient care optimized. The hospital sleep medicine team evaluated patients for home non-invasive ventilation close to their discharge date. Patients and their family members were provided education on the device by respiratory therapists and hospital-affiliated durable medical equipment staff before discharge. An appointment with the sleep medicine clinic run by the hospital sleep medicine service's advanced practice providers was provided for continuity of care post-discharge. A detailed description of the hospital sleep medicine service has been described by us previously [16]. Patients with prior diagnosis of acute-on-chronic hypercapnic respiratory failure and who were currently on home non-invasive ventilation were excluded. Data was extracted from the electronic medical record and REDCAP sleep medicine registry. Approval was obtained from the West Virginia University (WVU) institutional review board. (IRB # 2009120369). Consent was not obtained as data was collected retrospectively and analyzed anonymously. The data was accessed 09/10/2023.

The study cohort included patients with acute-on-chronic hypercapnic respiratory failure, irrespective of the etiology (i.e., COPD, obesity hypoventilation syndrome, restrictive lung disease). The patients were admitted to our hospital between January 2020 and July 2023 and received inpatient non-invasive ventilation. The inclusion criteria included the following: known history of hypercapnia by prior arterial blood gas showing $PaCO_2$ of >45mmHg; venous blood gas showing $PaCO_2$ >50 mmHg; elevated bicarbonate level (>27 mmol/L) without any other known etiology (e.g., diuretic use); documentation of acute-on-chronic hypercapnia on admission with pH < 7.35 and $Pco_2$ > 45 mmHg on arterial blood gas; and use of non-invasive ventilation during hospitalization. Qualifying criteria for post-discharge home non-invasive ventilation was recurrent readmission to the hospital for acute-on-chronic hypercapnia (50/95 patients) requiring in-patient non-invasive ventilation, or severe hypercapnic respiratory failure with inability to wean off non-invasive ventilation during hospitalization (described as acute elevation of $CO_2$ (reduction in PH by 0.05 or more) on a single night of non-use (45/90 patients). Most commonly, non-invasive ventilation patients were discharged on bilevel positive airway pressure mode (BiPAP), or average volume-assured pressure support therapy (AVAPS).

Adherence data was obtained for this cohort from the durable medical equipment companies and classified as either compliant or non-compliant with the post-discharge home non-invasive ventilation therapy. Adherence was defined as the use of non-invasive ventilation for ≥ 4 hours per night for at least 70% of the nights over 4 weeks, as measured in the first 60–90 days of initiation of the therapy. Information on patient reported barriers was obtained from clinic notes and encounters. OHS was defined as BMI > 30 Kg/m2, hypercapnia (arterial blood gas or venous blood gas) presence of sleep-disordered breathing (defined as obstructive sleep apnea or nocturnal; hypoxemia), and the absence of other causes of hypercapnia. Additionally, we collected data on pulmonary function testing, spirometry, and CT imaging findings of emphysema to differentiate COPD versus obesity hypoventilation syndrome patients.

The control group consisted of hospitalized patients who had hypercapnia ($PaCO_2$ ≥ 45 mmHg) on arterial blood gas or a $VPaCO_2$ ≥ 51 mmHg from their venous blood gas, comorbidity of COPD or a BMI ≥ 30 Kg/m2 who were admitted during the same time period but the primary admitting team did not consult hospital sleep medicine service, and those who did not qualify for non-invasive ventilation at discharge or experienced insurance denial.

The primary study outcomes were all-cause 12-month mortality, all-cause 6-month hospital readmissions, and emergency department visits. In the power analysis using a two-sided

test at a significance level of 0.05, this study with a total of 200 patients achieved 80% power to detect the difference in 12-month mortality from 1% or less for adherent patients to 12% or higher (non-adherent).

## Statistical analysis

Descriptive statistical analyses were performed to summarize the patient data and to describe the clinical outcomes, including summary tables, proportions, median, means, and standard deviations. The chi-square test and Fisher exact test were used to assess the association between categorical variables and the outcome variables, including hospital readmissions and mortality, whereas the Wilcoxon rank test was used to assess the difference of continuous variables between different groups. Kaplan-Meier method was used to estimate the survival curves, and the log-rank test was used to assess the difference of survival curves between different groups.

The hazard ratio (HR) and its 95% confidence interval were estimated using the Cox proportional hazards models, adjusting for potential confounding variables. Generalized linear regression models were used to assess ED visits and readmissions between the groups, adjusting for confounding variables. The Akaike information criterion was used to identify the final model, including independent predictors only. Analyses were considered statistically significant if $P < 0.05$. All statistical analyses were performed using statistical R software (R Foundation for Statistical Computing, Vienna, Austria; URL http://www.R-project.org/)

## Results

Of the 109 patients evaluated by the hospital sleep medicine service who also qualified for a post-discharge home non-invasive ventilation, 45% had COPD, 47% had OHS, and 8% had restrictive lung disease. Of these 109 patients, 14 were excluded from the analysis due to being deceased, discharged to hospice care, or not getting approved for non-invasive ventilation by their insurance carrier. Of the remaining 95 patients, 25 (26%) were found to be adherent with the post-discharge home non-invasive ventilation therapy at home, while 70 (64%) were not. A control group of 97 patients was also chosen based on the inclusion criteria (Fig 1). Pulmonary function test data was available for 41 of the 95 patients (43%). Of the control patients, 73

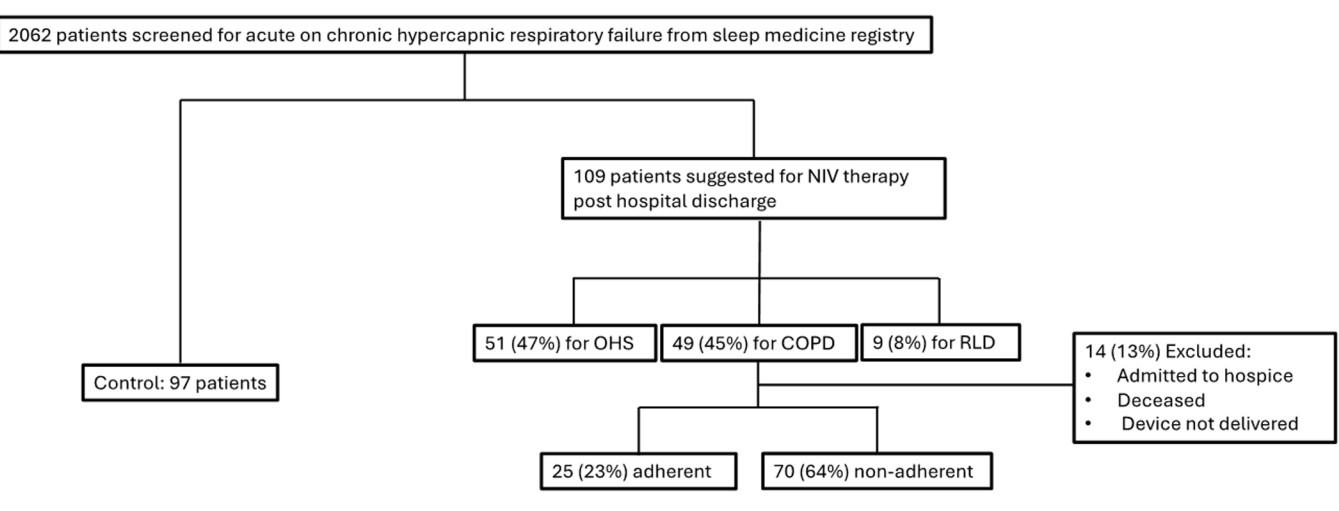

**Fig 1. Flowchart for the study's patient selection.**

(75%) were diagnosed with a comorbidity of COPD, of which 27 (37%) had pulmonary function test data available. The mean FEV1/FVC ratio % for the adherent, non-adherent, and control groups were 54.2 ± 15.8, 62.7 ± 22.0, and 69.5 ± 15.7, respectively. The FEV1% for the adherent, non-adherent, and control groups were 48.8 ± 33.0, 45.4 ± 16.9, and 59.3 ± 17.2, respectively.

## Baseline characteristics of the study sample

The mean age was 52.1± 16.7, 58.7±12.9, and 61.4±13.1 for the adherent, non-adherent, and control groups respectively. There was an equal proportion of males and females in these groups except in the adherent group which had more females than males (60%). Similarly, the mean BMI was 54.3±19.1, 46.4 (±17.4), and 43.4 (±14.8) in the adherent, non-adherent, and control groups respectively. More than 50% of the patients in all groups were smokers. Patient comorbidities were quite similar in all groups. Of note, approximately half of the patients in each group had baseline obstructive sleep apnea.

## Mortality

The adherent group had significantly lower mortality as compared to the non-adherent as well as the control group (p=0.022, log-rank test in Fig 2). Survival benefit persisted on multivariate analysis with the Cox regression model adjusting for comorbidities and demographics (HR = 0.05, p=0.04 in Table 2). The mortality among the non-invasive ventilation -adherent was 0% versus 11% in the NIV non-adherent versus 19% in the control group (HR= 0.05 versus 0.08 versus 1, (p=0.04 in Cox model).

## Hospital readmissions and ED visits

A similar trend was noted in the all cause 6-month hospital readmission and ED visits. There was a 52% 6-month readmission rate with the adherent group, 66% with the non-adherent group, and

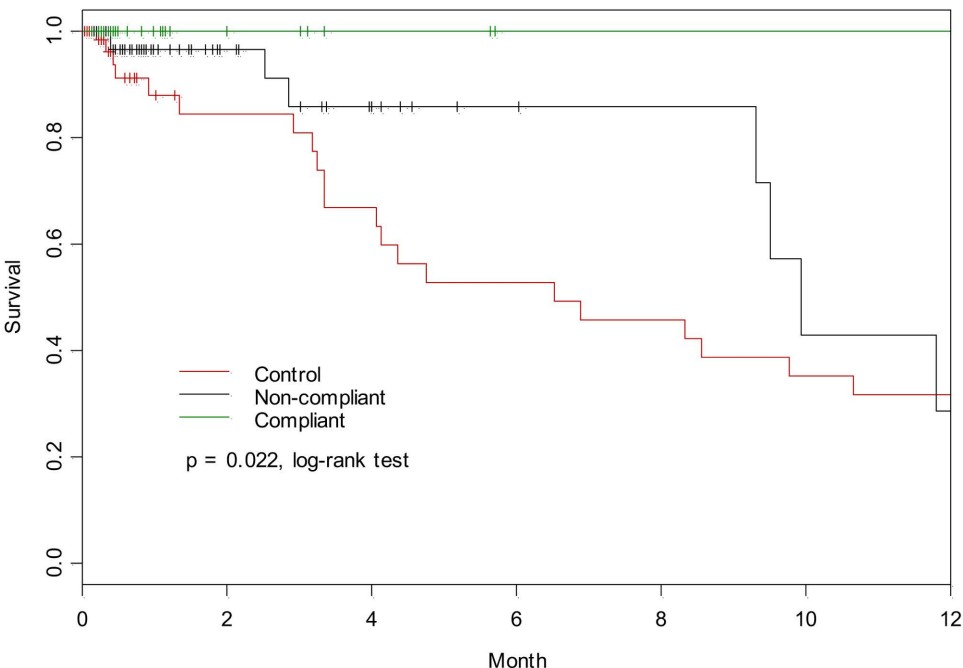

**Fig 2. Kaplan Meir curve showing 12-month survival between the adherent, non-adherent and control group.**

**Table 1. Patient characteristics, comorbidities and survival analysis. Hazard ratios were estimated from the fitted Cox proportional hazard models on mortality.**

| Characteristic | PDH-NIV adherent patients | PDH-NIV non-adherent patients | Control patients | Hazard Ratio (95% CI) | p-value |
|---|---|---|---|---|---|
| Demographics | | | | | |
| Participants, No. | 25 | 70 | 97 | | |
| Age, mean (SD): | 52.1 (±16.7) | 58.7 (±12.9) | 61.4 (±13.2) | 1.02 (0.97-1.07) | 0.38 |
| Female Sex, No. (%): | 15 (60%) | 38 (54%) | 48 (49.5%) | 0.99 (0.32-3.05) | 0.98 |
| BMI, mean (SD): kg/m^2 | 54.3 (±19.1) | 46.4 (±17.4) | 43.4 (±14.8) | 1.01 (0.96-1.05) | 0.79 |
| Neck Circumference, mean (SD): cm | 19.8 (±3.34) | 18.7 (±2.8) | | | |
| Comorbidities | | | | | |
| HTN, No. (%): | 14 (56%) | 56 (80%) | 81 (84%) | 0.39 (0.06-2.30) | 0.3 |
| CHF, No. (%): | 10 (40%) | 36 (51%) | 51 (53%) | 0.55 (0.15-1.99) | 0.37 |
| Atrial Fibrillation, No. (%): | 4 (16%) | 23 (33%) | 34 (35%) | 3.67 (0.99-13.51) | 0.051 |
| COPD/Asthma, No. (%): | 16 (64%) | 46 (66%) | 73 (75%) | 0.76 (0.21-2.79) | 0.68 |
| Diabetes, No. (%): | 10 (40%) | 38 (54%) | 55 (57%) | 1.01 (0.26-3.91) | 0.99 |
| OSA, No. (%): | 16 (64%) | 38 (54%) | 30 (31%) | 1.93 (0.56-6.66) | 0.3 |
| Cardiomyopathy, No. (%): | 1 (4%) | 7 (10%) | 17 (18%) | 0.55 (0.12-2.43) | 0.43 |
| Lung Emphysema, No. (%): | 6 (24%) | 3 (4%) | 13 (13%) | 0.46 (0.13-1.69) | 0.25 |
| CAD, No. (%): | 13 (52%) | 19 (27%) | 45 (46%) | 2.52 (0.80-7.89) | 0.11 |
| Stroke, No. (%): | 1 (4%) | 6 (9%) | 13 (13%) | 0.65 (0.17-2.51) | 0.53 |
| Smoking History, No. (%): | 13 (52%) | 45 (64%) | 61 (63%) | 2.72 (0.69-10.66) | 0.15 |
| Medications | | | | | |
| Beta Blockers, No. (%): | 13 (52%) | 46 (66%) | 62 (64%) | 3.77 (1.11-12.85) | 0.034 |
| ACE Inhibitors, No. (%): | 6 (24%) | 16 (23%) | 24 (25%) | 1.23 (0.36-4.26) | 0.74 |
| ARBs, No. (%): | 4 (16%) | 9 (13%) | 18 (19%) | 3.85 (0.70-21.27) | 0.12 |
| Inhaled Bronchodilators, No. (%): | 19 (76%) | 52 (74%) | 63 (65%) | 0.74 (0.20-2.75) | 0.65 |
| Diuretics, No. (%): | 19 (76%) | 53 (76%) | 59 (61%) | 0.77 (0.21-2.80) | 0.69 |
| PaCO2, mean (SD): | 75.4 (±15.0) | 69.1 (±26.1) | 60.0 (±14.3) | 1.02 (0.99-1.05) | 0.14 |
| ODI, mean (SD) | 10.8 (±14.8) | 12.8(±16.5) | 12.7(±18.6) | 0.99 (0.98,1.02) | 0.876 |
| No Show rate, %, Mean (SD) | 7.5 (±5.5) | 9.1(±7.1) | 9.7(±10.4) | 0.97 (0.92, 1.02) | 0.515 |

**Table 2. Hazard ratio comparison between adherent patients, non-adherent patients, and controls, based on the multivariable Cox model on mortality adjusting for patient characteristics given in the previous table.**

| | Coefficient | S.E. | Hazard ratio | low95% | high95% | p-value |
|---|---|---|---|---|---|---|
| Control | ref | ref | 1.00 | ref | ref | ref |
| non-adherent | -1.09 | 0.64 | 0.34 | 0.10 | 1.18 | 0.088 |
| Adherent | -2.92 | 1.42 | 0.05 | 0.00 | 0.87 | 0.04 |

65% with the control group. The 6-month ED visit noted 12%, 24%, and 21% rates with adherent, non-adherent, and control groups, respectively. In the multivariate analysis of 6-month ED visits, a generalized linear model was used to assess the number of incidences, adjusting for potential confounding variables. The final model was identified based on the backward variable selection according to Akaike information criterion. The result showed that the incidence rate ratio for ED visits was 0.62 (95% CI: 0.39–0.99, p=0.049) between adherent and non-adherent groups, and 0.33 (95% CI: 0.12–0.86, p=0.024) between adherent and control groups.

## Reasons for non-compliance

Of the total group of 70 non-compliant patients, 25 (36%) were found to be intolerant to mask or have adverse effects, 19 (27%) were found to have poor understanding/low health

**Table 3. Tabulation of reasons for non-compliance\*.**

| Reasons for Non-Compliance (n=70) | |
|---|---|
| Intolerant (25) | ■ Did not meet the criteria for compliance (≥4 hours per night for at least 70% of days)<br>■ Mask leak due to poor fit<br>■ Adverse effects (e.g. headache, nasal congestion, facial abrasions from the mask, claustrophobia)<br>■ Patient comorbidities restricting use (e.g. neck pain from spinal degenerative joint disease) |
| Low Health Literacy (19) | ■ Determined to have a Health Literacy Score of ≤18 based on WVU Medicine Health Literacy Questionnaire (≤18 regarded as low literacy)<br>■ Clinical judgement from providers who evaluated patients for device |
| Discharged to another facility (Skilled Nursing or Inpatient Rehabilitation (17) | ■ Utilized skilled nursing facility devices so compliance reports could not be obtained<br>■ Inpatient Rehabilitation facilities sent orders to DME companies, so unclear if patients got the device/ inability to obtain usage information |
| Patient Refusal (3) | ■ Patient did not receive device due to patient refusal |
| Miscellaneous (6) | ■ Unable to obtain report<br>■ Patients did not return the Secure digital (SD) card to DME for obtaining reports<br>■ Patients admitted to outside healthcare system<br>■ Financial burden |

\* Did not meet the criteria for compliance (≥4 hours per night for at least 70% of days).

literacy score, and 17 (24%) were discharged to a skilled nursing home/rehabilitation center. The other reasons were insurance denial and miscellaneous (Table 3). As a measure of general compliance, we included "no-show" rate statistics, which is a measure of adherence for any clinic visit/follow ups and testing (blood test, imaging, and elective procedures were reported in the electronic medical record as a percent).

## Discussion

Acute-on-chronic respiratory failure at admission may indicate patients who are at risk for unfavorable short- and long-term outcomes [5]. In this single-center, cross-sectional study, we found that adherence to non-invasive ventilation following hospital discharge for acute-on-chronic respiratory failure is associated with lower overall mortality and fewer ED visits within a rural cohort. In our study, the overall mortality rate was 4% in the non-invasive ventilation-adherent group (HR 0.05, p=0.04) and 14% in the non-adherent group (HR 0.34, p=0.088) after adjusting for demographics and comorbidities at 12 months, whereas the control group's mortality rate was 3%. The study also showed an 82.3% reduction in ED visits in the non-invasive ventilation-adherent group compared to the non-adherent (3% vs. 17%; p=0.049) and an 88% reduction compared to controls (3% vs. 25%; p=0.024) at 6 months. Although a downward trend in readmissions was noted in the adherent group compared to the control, this difference was not statistically significant.

This study is the first to demonstrate that adherence to post-discharge non-invasive ventilation in patients admitted for acute-on-chronic hypercapnic respiratory failure from mixed etiologies may improve clinical outcomes in a rural population. This finding is particularly significant when considering the healthcare utilization and healthcare access disparities in these communities. Rural Appalachia faces a higher prevalence of COPD, obesity hypoventilation syndrome, and restrictive lung disorders due to a shortage of primary care physicians

and pulmonary specialists. The region's harsh geographics also allows for only a limited access to ambulatory healthcare. As a result, many rural patients depend on emergency departments and hospitals for their healthcare needs [17,18]. Given that hospital admissions frequently represent the primary healthcare interaction for these vulnerable communities, implementing hospital practice models that enhance access and adherence to non-invasive ventilation around the time of discharge may help decrease subsequent emergency department visits, hospital readmissions, and mortality following an acute-on-chronic hypercapnic respiratory exacerbation.

Our results are consistent with a number of randomized and comparative studies assessing the impact of hospital discharge with empirical non-invasive ventilation on mortality in hospitalized patients with particular hypercapnic respiratory failure illnesses following an acute exacerbation [2,19,20,24]. For example, observational studies have shown that patients with obesity hypoventilation syndrome discharged with non-invasive ventilation after an acute-on-chronic hypercapnic exacerbation experienced lower mortality rates up to 1 year of follow-up [2,19]. In a meta-analysis study, Mokhlesi and colleagues have shown that hospital discharge with positive airway pressure reduces mortality at 3 months following acute-on-chronic hypercapnic respiratory failure in patients with OHS or suspected of having obesity hypoventilation syndrome (relative risk 0.12; 95% CI, 0.05–0.30, risk difference -14.5%) [2,19]. Similar responses have been noted in other studies on obesity hypoventilation syndrome, having significantly lower mortality with long term non-invasive ventilation compared to those who remain untreated [2,20]. Obesity hypoventilation syndrome is an important etiology of acute-on-chronic hypercapnic respiratory failure in Appalachia due to the high prevalence of obesity. This is highlighted by the fact that the mean BMI in our cohort of patient-prescribed non-invasive ventilation was 48.5 (SD 18.14) compared to a similar study in an urban environment, which was 33.7 (SD 12) [21].

Rural obesity is a major healthcare problem, and it is estimated that the prevalence of obesity in rural areas is 6.2 times more than those of urban areas [22]. Similarly, Murphy and colleagues demonstrated that home non-invasive ventilation provided to patients with COPD and persistent hypercapnia 2–4 weeks after hospital discharge was associated with a reduced risk of both readmission and mortality within 1 year (absolute risk reduction 17.0%; 95% CI, 0.1–34.0%) [23]. Although home non-invasive ventilation was initiated in an outpatient setting, participants received in-hospital acclimation and titration. However, unlike previous research, our study provides new information by analyzing a cohort of patients in a rural setting. Our cohort had acute-on-chronic hypercapnic respiratory failure due to a variety of etiologies applying the usual clinical practice for accessing non-invasive ventilation modes at the time of discharge under Medicare/Medicaid regulations.

In our study, the increased access to non-invasive ventilation therapy upon discharge and at follow-up was facilitated by a well-established inpatient clinical pathway previously described in prior publications [16]. This practice incorporates an integrated hospital-based sleep medicine model, tailored to serve a population with predominately Medicare/Medicaid coverage. This model ensures patients receive the necessary support, education, and resources for non-invasive ventilation therapy upon discharge and follow-up. When a hospitalized patient is assessed for being at high risk for adverse outcomes—such as hypercapnia without positive airway pressure therapy—the hospital sleep medicine team works with the insurance and local durable medical equipment company to get the patient approved for post-discharge non-invasive ventilation. In a few instances, a loaner non-invasive ventilation device is provided by the local durable medical equipment company. The patient returns or exchanges these loaner devices following a confirmatory study conducted after discharge. Our sleep

center has a dedicated bed available for the urgent accommodation of patients within 72 hours of hospital discharge.

In other sleep-related breathing disorders, adherence to positive airway pressure therapy has been shown to reduce annual healthcare-related expenses by 40% among Medicare patients [24]. Our group showed a reduction in readmission, ED visits, and health care cost savings in hospitalized patients proactively screened and initiated on positive airway pressure therapy [25]. Therefore, identifying modifiable factors impacting adherence may reduce healthcare costs. It should also be highlighted that our control group had the worst outcome. It is possible that the non-adherent group's limited usage was better than no therapy. Several studies in the past have documented incremental benefits with increased hourly usage [26,27]. We used strict Centers for Medicare & Medicaid Services criteria for defining adherence (4 hours of nightly use, and 70% or more of nights over a minimum period of 4 weeks). Hence, the non-adherent group included patients with partial adherence. Our observational study not only underscores the impact of adherence to non-invasive ventilation therapy on clinically significant outcomes but also identifies common causes of noncompliance, such as treatment barriers (e.g., poor mask fit) and low health literacy based on a low West Virginia University (WVU) Medicine literacy score.

Health literacy is an individual's ability to obtain, process, and understand basic health information and services necessary for making appropriate health decisions, and it appears to be an independent predictor of treatment success [28,29]. Health literacy may represent a potentially important modifiable risk factor for poor treatment outcomes in this population, as low scores—which are more common among rural populations than their urban counterparts—have been shown to reduce adherence to medications [17,28,29]. This may be an important area of focus to improve adherence to non-invasive ventilation and subsequent outcomes in rural areas, which is also consistent with the goals of 2030 Healthy People public health priorities [30]. Elsewhere, discharge to long term facilities also represents an area where the transition of care can be strengthened.

Our study has several strengths. It was a rigorously designed observational study describing a cohort of rural patients with mixed etiologies of acute-on-chronic hypercapnic respiratory failure discharged with non-invasive ventilation versus those discharged without it. We thoroughly reviewed medical records and excluded participants with advanced conditions needing palliative care, where life expectancy was less than 1 year. We further compared no-show rates in all 3 groups as a surrogate marker for general compliance (Table 1). The study also implemented a well-validated clinical pathway by a hospital-based sleep medicine consultation practice, ensuring a consistent and systematic approach to qualifying patients for non-invasive ventilation and providing them with home devices at discharge and follow-up. Additionally, we analyzed objective adherence data based on download reports following hospital discharge, which enhances the robustness of our findings.

This study has several limitations. Being a single-center investigation, its findings may not be widely generalizable. However, this design enhances internal validity due to consistent data abstraction of clinical information and mortality from a unified regional electronic medical record system. Although the rural setting of our study may limit its applicability to other rural regions and urban environments, the successful implementation of a previously proven clinical pathway by a hospital-based sleep medicine team suggests that this approach could be extrapolated to other medical practices across the country. Additionally, we could not determine specific causes of mortality, nor could we confirm whether patients discharged without positive airway pressure therapy eventually received it within the year, although we suspect it is unlikely. The study also coincided with the covid-19 pandemic and its impact remains

uncertain. Finally, as an observational study, we cannot attribute causality or exclude the role of unmeasured confounders in these outcomes.

In summary, hospitalized patients with acute-on-chronic hypercapnic respiratory failure in a rural region who adhere to non-invasive ventilation after discharge are associated with better survival rates and fewer ER visits. Based on this observational evidence, we advise providing empiric non-invasive ventilation for home use to patients hospitalized for acute-on-chronic hypercapnic respiratory failure and subsequent follow-up to ensure adherence. However, randomized controlled trials are necessary to address the critical limitations of this and other observational studies and to more accurately assess whether hospitalized patients with acute-on-chronic hypercapnic respiratory failure should be discharged on non-invasive ventilation therapy. These studies should focus on mortality, the cost-effectiveness of discharging patients on empiric non-invasive ventilation, and patient-centered outcomes such as quality of life.

## Conclusion

This cross-sectional study outlines the disparity in the utilization of post-discharge home non-invasive ventilation therapy in our rural population with chronic hypercapnic respiratory failure, as well as notable reductions in mortality, hospital readmissions, and ED visits with therapy adherence. There is a need to understand the unique barriers that the rural population may experience to impact adherence. Further work to expand on our findings, including quantification of the barriers to non-invasive ventilation adherence and implementation of new healthcare pathways for enhancing the use of post-discharge home non-invasive ventilation therapy in rural communities of the United States, is recommended.

## Author contributions

**Conceptualization:** Sunil Sharma, Robert Stansbury, Mayuri Mudgal, Priyanka Srinivasan, Edward Rojas, Kassandra K. Olgers, Scott Knollinger, Sijin Wen.

**Data curation:** Sunil Sharma, Robert Stansbury, Mayuri Mudgal, Priyanka Srinivasan, Edward Rojas, Kassandra K. Olgers, Scott Knollinger.

**Formal analysis:** Sunil Sharma, Robert Stansbury, Edward Rojas, Kassandra K. Olgers, Bernardo J. Selim.

**Funding acquisition:** Sunil Sharma.

**Investigation:** Sunil Sharma, Robert Stansbury, Mayuri Mudgal, Priyanka Srinivasan, Kassandra K. Olgers, Scott Knollinger, Sijin Wen.

**Methodology:** Sunil Sharma, Robert Stansbury, Mayuri Mudgal, Priyanka Srinivasan, Edward Rojas, Kassandra K. Olgers, Scott Knollinger, Bernardo J. Selim, Sijin Wen.

**Project administration:** Sunil Sharma, Robert Stansbury, Mayuri Mudgal, Priyanka Srinivasan, Scott Knollinger.

**Resources:** Sunil Sharma, Sijin Wen.

**Software:** Sunil Sharma, Edward Rojas, Kassandra K. Olgers, Sijin Wen.

**Supervision:** Sunil Sharma, Robert Stansbury, Mayuri Mudgal, Priyanka Srinivasan, Edward Rojas, Kassandra K. Olgers, Scott Knollinger.

**Validation:** Sunil Sharma, Robert Stansbury, Mayuri Mudgal, Priyanka Srinivasan, Edward Rojas, Bernardo J. Selim, Sijin Wen.

**Visualization:** Sunil Sharma, Robert Stansbury, Mayuri Mudgal, Priyanka Srinivasan, Kassandra K. Olgers, Scott Knollinger, Bernardo J. Selim, Sijin Wen.

**Writing – original draft:** Sunil Sharma, Robert Stansbury, Mayuri Mudgal, Priyanka Srinivasan, Edward Rojas, Kassandra K. Olgers, Bernardo J. Selim, Sijin Wen.

**Writing – review & editing:** Sunil Sharma, Mayuri Mudgal, Priyanka Srinivasan, Edward Rojas, Kassandra K. Olgers, Scott Knollinger, Bernardo J. Selim, Sijin Wen.

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
