## [Decision Letter · Decision Letter 0]

4 Feb 2025

PONE-D-24-60120Post-Discharge Non-Invasive Ventilation for Hypercapnic Respiratory Failure: Outcomes in a Rural CohortPLOS ONE

Dear Dr. Sharma,

Thank you for submitting your manuscript to PLOS ONE. After careful consideration, we feel that it has merit but does not fully meet PLOS ONE’s publication criteria as it currently stands. Therefore, we invite you to submit a revised version of the manuscript that addresses the points raised during the review process.

Please revise the limitations section of the manuscript to address issues raised by Reviewer 2 below.In addition, please clarify some of the questions raised regarding methodology. 

We look forward to receiving your revised manuscript.

Kind regards,

Rabail Arif Chaudhry

Academic Editor

PLOS ONE

Journal Requirements:

2. Thank you for stating the following in the Acknowledgments Section of your manuscript: "Research reported in this publication was supported by the National Institute Of General Medical Sciences of the National Institutes of Health under Award Number 5U54GM104942-08. The content is solely the responsibility of the authors and does not necessarily represent the official views of the National Institutes of Health."

Please remove any funding-related text from the manuscript and let us know how you would like to update your Funding Statement. Currently, your Funding Statement reads as follows: “The authors received no specific funding for this work.”.

[SS declares receives grant and support for attending meetings or travel from West Virginia University (WVU) and National Science Foundation; is on speakers bureau of Zoll Respicardia Inc. Grant funding from INARI Medical Inc PEERLESS II trial, NIH RECOVER trial, rEST trial by Zoll Respicardia Inc.; and holds patents with WVU (invention numbers: US2023/0122156) that has been licensed by Premash Inc. SS is consultant for Premash Inc.

Dr. Stansbury reports grants from NIH. Dr. Rojas and Dr. Mudgal have no conflicts to declare. Priyanka Srinivasan, Kassandra Olgers, Scott Knollinger, and Wen Sijin also have no conflicts to declare].

We note that you received funding from a commercial source: [Premash Inc and Zoll Respicardia Inc.]

Within this Competing Interests Statement, please confirm that this does not alter your adherence to all PLOS ONE policies on sharing data and materials by including the following statement: ""This does not alter our adherence to PLOS ONE policies on sharing data and materials.” (as detailed online in our guide for authors http://journals.plos.org/plosone/s/competing-interests ). If there are restrictions on sharing of data and/or materials, please state these. Please note that we cannot proceed with consideration of your article until this information has been declared.

Reviewers' comments:

Reviewer's Responses to Questions

**Comments to the Author**

1. Is the manuscript technically sound, and do the data support the conclusions?

Reviewer #1: Yes

Reviewer #2: Yes

2. Has the statistical analysis been performed appropriately and rigorously? 

Reviewer #1: Yes

Reviewer #2: Yes

3. Have the authors made all data underlying the findings in their manuscript fully available?

Reviewer #1: No

Reviewer #2: Yes

4. Is the manuscript presented in an intelligible fashion and written in standard English?

Reviewer #1: Yes

Reviewer #2: Yes

5. Review Comments to the Author

Reviewer #1: I would have liked more explanation on the randomization process- since the control group consisted of participants who "did not qualify for non-invasive ventilation at discharge," but the study group did not include that subset of patients then doesn't that naturally skew your results? Are the patients first randomized into control vs study groups and then determined if they met criteria for NIV at discharge? That way, there would naturally be patients in both control and study groups that did not end up meeting criteria for NIV at discharge and their data removed for purposes of this study.

I would also have liked a breakdown for chief complaint causing the 12 month mortality and hospital readmissions out of curiosity to see what percentage were pulmonary related.

Reviewer #2: The article is technically sound but has notable minor methodological limitations.

Patients with a prior diagnosis of acute-on-chronic hypercapnic respiratory failure who were already on home NIV were excluded. This exclusion may bias the study population toward newly diagnosed or less severe cases, potentially limiting the applicability of the findings to patients with long-standing NIV use or more advanced disease.

The retrospective data, relied on data extracted from electronic medical records and a sleep medicine registry and hence prone to selection bias, missing data, and inaccuracies in documentation. Adherence data obtained from durable medical equipment companies, may not capture the full picture of patient usage patterns and may not account for patient-reported barriers (e.g., discomfort, mask fit issues) or technical issues with the devices. Elaborating on the methods these equipment companies use to collect adherence data may provide helpful context for readers.

Lastly, the study included patients admitted between January 2020 and July 2023, i.e. during the COVID-19 pandemic which most likely may have influenced hospitalization patterns, access to care, and outcomes, how were these factors addressed and adjusted for in the study?

6. PLOS authors have the option to publish the peer review history of their article (what does this mean? ). If published, this will include your full peer review and any attached files.

**Do you want your identity to be public for this peer review?** For information about this choice, including consent withdrawal, please see our Privacy Policy .

Reviewer #1: **Yes: ** Ronald Tang

Reviewer #2: **Yes: ** Maaz Shah Khan

---

## [Author Response · Author response to Decision Letter 0]

28 Feb 2025

Response to reviewers comments for NIV study

Editorial comments

Please remove any funding-related text from the manuscript and let us know how you would like to update your Funding Statement. Currently, your Funding Statement reads as follows: “The authors received no specific funding for this work.”.

Funding related text removed from the manuscript as advised.

Please add the following funding information on the submission form:

“Research reported in this publication was supported by the National Institute Of General Medical Sciences of the National Institutes of Health under Award Number 5U54GM104942-08. The content is solely the responsibility of the authors and does not necessarily represent the official views of the National Institutes of Health.”

Thank you.

Amended competing interests statement added to the manuscript and added the sentence “This does not alter our adherence to PLOS ONE policies on sharing data and materials.”

Please review your reference list to ensure that it is complete and correct.

Reference list is complete and correct.

5. Review Comments to the Author

Reviewer #1: I would have liked more explanation on the randomization process- since the control group consisted of participants who "did not qualify for non-invasive ventilation at discharge," but the study group did not include that subset of patients then doesn't that naturally skew your results? Are the patients first randomized into control vs study groups and then determined if they met criteria for NIV at discharge? That way, there would naturally be patients in both control and study groups that did not end up meeting criteria for NIV at discharge and their data removed for purposes of this study.

We appreciate the reviewer’s observation. Most patients in the control group were admitted by the primary team for acute-on-chronic hypercapnic respiratory failure and were not evaluated by the hospital sleep medicine service. Additionally, we included patients who were consulted but were unable to be discharged on NIV. This approach was appropriate, as the study aimed to assess whether patients who were not discharged on therapy were at higher risk for poor outcomes. As noted in Table 1, there were no significant differences between the intervention and control groups. Since this is a retrospective study based on a prospective registry, the control group was not initially randomized. To clarify, we have updated the methodology as follows:

“The control group consisted of hospitalized patients who had hypercapnia (PaCO2 ≥ 45 mmHg) on arterial blood gas or a VPaCO2 ≥ 51 mmHg from their venous blood gas, comorbidity of COPD or a BMI ≥ 30 Kg/m2 who were admitted during the same time period but the primary admitting team did not consult hospital sleep medicine service, and those who did not qualify for non-invasive ventilation at discharge or experienced insurance denial.”

I would also have liked a breakdown for chief complaint causing the 12 month mortality and hospital readmissions out of curiosity to see what percentage were pulmonary related.

Unfortunately, we could not determine specific causes of mortality as records from outside hospitals were not available. This has been listed in the limitations section. Prospective studies in on further delineation of causes for mortality are recommended.

Reviewer #2: The article is technically sound but has notable minor methodological limitations.

Patients with a prior diagnosis of acute-on-chronic hypercapnic respiratory failure who were already on home NIV were excluded. This exclusion may bias the study population toward newly diagnosed or less severe cases, potentially limiting the applicability of the findings to patients with long-standing NIV use or more advanced disease.

We thank the reviewer for their excellent comments.

We believe it was important to exclude patient already on NIV as the duration of use and hours of compliance could not be controlled. We agree with the reviewer that enrolling NIV naïve patients may have biased the cohort towards less severe cases. However our data shows that NIV was beneficial even in relatively less severe patients and supports the conclusion that more severe patients may also benefit.

The retrospective data, relied on data extracted from electronic medical records and a sleep medicine registry and hence prone to selection bias, missing data, and inaccuracies in documentation. Adherence data obtained from durable medical equipment companies, may not capture the full picture of patient usage patterns and may not account for patient-reported barriers (e.g., discomfort, mask fit issues) or technical issues with the devices. Elaborating on the methods these equipment companies use to collect adherence data may provide helpful context for readers.

We again thank the reviewer for their comments and agree with the observations. We have addressed this in the limitations section. Downloadable objective adherence data from the DME company was obtained and reviewed by the investigators and was not based solely on DME companies observations or documentation. Information on patient reported barriers was obtained from clinic notes and encounters. This has been added to the methodology section.

Lastly, the study included patients admitted between January 2020 and July 2023, i.e. during the COVID-19 pandemic which most likely may have influenced hospitalization patterns, access to care, and outcomes, how were these factors addressed and adjusted for in the study?

We agree with the reviewer that the COVID-19 pandemic affected all aspects of medical care, and the impact of these confounders is difficult to quantify. However, regarding acute-on-chronic hypercapnic respiratory failure, we did not observe a decrease in cases but rather a slight increase, likely due to the temporary closure of ambulatory clinics. Both the intervention and control groups were exposed to similar environmental influences. We have acknowledged the unknown impact of COVID-19 in the study’s limitations section.

Done

---

## [Editor Report · Decision Letter 1]

7 Mar 2025

Post-Discharge Non-Invasive Ventilation for Hypercapnic Respiratory Failure: Outcomes in a Rural Cohort

PONE-D-24-60120R1

Dear Dr. Sharma,

We’re pleased to inform you that your manuscript has been judged scientifically suitable for publication and will be formally accepted for publication once it meets all outstanding technical requirements.

Kind regards,

Rabail Chaudhry

Academic Editor

PLOS ONE
---

## [Editor Report · Acceptance letter]

PONE-D-24-60120R1

PLOS ONE

Dear Dr. Sharma,

I'm pleased to inform you that your manuscript has been deemed suitable for publication in PLOS ONE. Congratulations! Your manuscript is now being handed over to our production team.

Kind regards,

on behalf of

Dr. Rabail Arif Chaudhry

Academic Editor

PLOS ONE